# Calibrating Prompt from History for Continual Vision-Language Retrieval and Grounding

## ABSTRACT

In the field of machine learning, continual learning is a crucial concept that allows models to adapt to non-stationary data distributions. However, most of the existing works focus on uni-modal settings and ignore the multi-modal data. In this paper, to enable neural networks better understand diverse modalities in real-world scenario, we investigate continual learning for two typical vision-language applications, i.e. retrieval and grounding. Instead of conventional exemplar-based methods, we leverage the pre-trained transformer model (e.g. CLIP/GLIP) and the prompt technique to tackle this problem. Under this scheme, we identify two critical limitations in existing methods: (1) Unfamiliarity across tasks, which prevents task-specific prompts from achieving forward propagation; and (2) Heterogeneity between modalities, which makes it difficult to guarantee a consistent optimization direction for prompts of different modalities. To overcome these constraints, we design Historical Prompt Calibration that includes two objectives to calibrate prompts. First, the intra-modal relevance estimation helps encode sufficient task-specific information for prompts, with the help a relevance estimator developed for recognizing task relevance. Second, the inter-modal consistency alignment enhances the agreement of the two modality-specific prompts in the current task by contrasting them with the prompts from previous tasks. We evaluate the superiority of our strategy over state-of-the arts methods by four vision-language applications, including two retrieval tasks (i.e. image- and video-text retrieval) and two grounding tasks (i.e. referring expression comprehension and segmentation).

## CCS CONCEPTS

• **Information systems → Multimedia and multimodal retrieval**.

## KEYWORDS

Continual learning, multi-modal, prompt, retrieval, grounding

## 1 INTRODUCTION

Continual learning, the ability to learn sequentially from a continuous data stream, is a fundamental requirement for intelligent systems to work effectively in the real world. Numerous methods have been proposed to tackle this problem. Regularization-based methods [1, 2, 22, 55] regularize key parameters during continual learning, which is simple but limited in some challenging applications. Replay-based and architecture-based methods are more effective, which save a small set of examples in a memory [7, 33, 41, 43, 46] and learn separated parameters for each task [17, 23, 53] respectively. While promising, these two methods are still limited due to the requirement for extra memory space and large amount of additional parameters. Recently, prompt-based methods [12, 40, 49, 50] has gained popularity, which adapt the pre-trained models to continuous tasks by learning task-specific prompt tokens for each individual task. This method has been demonstrated to be highly effective with minimal parameter tuning, leading to increased interest and research in this field.

Despite the remarkable progress made in addressing the continual learning problem, the majority of existing works have been limited to uni-modal tasks such as image classification [50] and fake detection [48]. This is a significant drawback in modern society where multi-modal data is ubiquitous, spanning a range of modalities such as vision, language, sound, and more. Therefore, it is imperative to expand the current research in continual learning to support the diverse modalities.

In this paper, we strive to advance deep learning in real-world scenarios by tackling the multi-modal challenge in continual learning. We focus on two major and common modalities, i.e. vision and language, and study the continual learning in two representative applications: retrieval and grounding. In light of the efficiency of prompt technique as well as the recent success of multi-modal pre-training models, we can seamlessly follow the prompt-based pipeline to develop our research.

However, existing prompt-based methods have limitations for continual learning and incorporating another modality poses more difficulty. As illustrated in Figure 1, we observe two inevitable challenges: (1) Unfamiliarity across tasks. Before diving into the multi-modal property, we note previous studies [48, 50] merely focus on avoiding *catastrophic forgetting* while ignore the *forward transfer*. They typically learn independent prompts for each task, which fails to leverage the knowledge from past tasks for efficient learning of new tasks. (2) Heterogeneity between modalities. The two-stream architecture [25, 38, 44] is a typical vision-language transformer model that applies two separate transformers to encode each modality. Directly learning modality-specific prompt for each encoder may not ensure the consistent optimization for these prompts.

Based on the aforementioned discussions, we propose Historical Prompt Calibration, a novel strategy to explicitly calibrate task-wise and modality-wise prompt relation by learning from history knowledge. First, to alleviate the task unfamiliarity, we design intra-modal relevance estimation that encourages prompts to be more closely related to the current task by leveraging prompts of past

*ACM MM, 2024, Melbourne, Australia*

© 2024 Copyright held by the owner/author(s). Publication rights licensed to ACM.
ACM ISBN 978-x-xxxx-xxxx-x/YY/MM
https://doi.org/10.1145/nnnnnnn.nnnnnnn

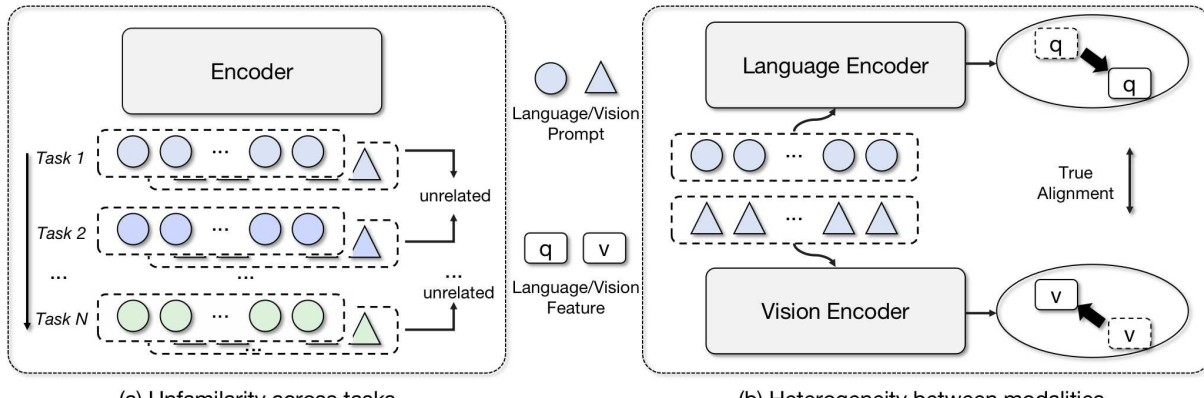

**Figure 1: Two critical challenges: Unfamiliarity across tasks and Heterogeneity between modalities.**

tasks. This technique involves developing a relevance estimator on the pre-trained model to discern whether the input prompt corresponds to the current task, where the positive samples are the learnable prompts of the current task and the negative samples are sampled from frozen prompts of the previous tasks. This objective help improves the quality of the prompt, because it explicitly forces each task-specific prompt group to be distinctive from others and hence enable them preserve much task-specific information. Second, to bridge the gap between modalities, we present inter-modal consistency alignment to enhance consistency between two modal prompts within the same task. This approach employs contrastive learning to ensure the agreement between two modal features that are encoded under prompt guidance of the current task can surpass all other combinations, where either modal feature is encoded by historical prompts. By similarly utlizing historical prompts in a cross-modal manner, we can enhance the cross-modal alignment to a greater extent.

To validate the effectiveness of our proposed approach, we conduct the extensive experiments on four important vision-language applications, including image-text retrieval, video-text retrieval, referring expression comprehension and segmentation. Our comprehensive evaluation reveals that our HPC approach can deliver state-of-the-art results, underscoring its impressive power.

## 2  RELATED WORKS

**Continual Learning.** Existing approaches to continual learning can be categorized into three main groups: (1) Replay-based methods [6, 7, 14, 20, 33, 41–43, 46] store a subset of data from previous tasks for future rehearsal via experience replay, representation consolidation or constrained optimization. The data can be either stored directly or synthesized by generative models.(2) Regularization-based methods [1, 2, 22, 55] enforce constraints on parameter changes to mitigate interference with prior tasks. (3) Architecture-based methods [17, 23, 53] learn separate sets of parameters dedicated to individual tasks. Despite their promises, replay-based methods suffer from diminishing performance with smaller buffers and raise data privacy concerns [5], the regularized methods struggle to achieve satisfactory performance in challenging settings [41, 51] and the architecture-based methods [19] require

substantial amount of additional parameters. Recently, the prompt-based methods are investigated [12, 50], which leverage learnable prompt parameters upon pre-trained models to encode knowledge more succinctly and hence avoid memory consumption. Following this paradigm, we establish our base framework upon S-Prompt [48] and further explore continual learning for vision-language retrieval and grounding.

**Visual-Language Transformer.** With the remarkable progress in language tasks [4, 39, 47], the transformer architecture is also being rapidly transferred to the field of computer vision [8, 10]. Recently, pretraining visual-language transformer [16, 25, 35, 37, 38, 45, 58] has yielded substantial improvements across various downstream tasks, e.g. visual question answering, image captioning [58], and referring image comprehension [25]. The large-scale models demonstrate extremely powerful capabilities of multi-modal pre-training. Among them, the single-stream architecture [16, 24, 37, 44] employs a single transformer to jointly model a pair of text and image, while the two-stream architecture [25, 38, 45] applies two transformers to separately learn the representations of the text and the image, respectively. In this work, we primarily leverage the two-stream architecture transformers as our modal encoders.

**Prompt Technique.** The prompt technique is initially applied in the NLP (Natural Language Processing) domain and subsequently adapted to vision and vision-language models. The fundamental concept behind prompt technique is to learn a function that modifies input texts or images, enabling language or image models to acquire additional task-related information [30]. A variety of prompting methods have emerged, including notable contributions such as [3, 12, 15, 18, 27, 50, 56, 57]. For example, [27] employs prefix-tuning to prompt the pre-trained language model. VPT [18] introduces a small number of learnable parameters into the input space of Vision Transformers (ViT) [11] and Swin Transformers (Swin) [32], outperforming full fine-tuning in many cases while also reducing storage costs. CoOp [56] incorporates prompts into the input of the vision encoder for the vision-language model. Recent work [48–50] exploit the prompt technique for continual learning in uni-modal applications. S-Prompts [48] introduces learnable prompts into the input space of both the visual and text encoder within a vision-language model. However, these methods fail to realize the task-wise and modality-wise connection for prompts, thereby falling

short in addressing the challenges of continuous vision-language retrieval and grounding.

# 3 HISTORICAL PROMPT CALIBRATION

## 3.1 Problem Formulation

Our research focus on the problem of continual learning for vision-language retrieval and grounding. In this problem, the model is required to learn knowledge from a series of tasks (i.e. input data), ultimately evolving into a universal expert capable of handling all tasks. Formally, the model learns tasks $T = \{T^1, T^2, ..., T^k\}$ in a sequential manner, where $K$ denotes the total number of tasks. Each task $T^k$ is associated with data $\mathbb{D}_k$ from different or even highly heterogeneous contents. At task $T^k$, the model receives the incoming data $\mathbb{D}_k = \left\{ q_i^k, v_i^k, a_i^k \right\}_{i=1}^{N_k}$ where $q_i^k, v_i^k$ is the $i$-th input language modality and visual modality (e.g. image/video) from task $k$ respectively, and $a_i^k$ is the corresponding label, (e.g. $a_i^k \in \{0, 1\}$ for image-text retrieval and $a_i^k = (x, y, h, w)$ for referring expression comprehension), and $N_k$ is the total number of samples at task $T^k$. In order to prioritize data privacy and minimize memory consumption, we adhere to the exemplar-free setting, which prohibits the utilization of previously seen data during both training and inference stages.

Conventional continual learning methods have long been classified into task-, class-, and domain-incremental settings according to various task transition environments. However, the domain of continual vision-language retrieval and grounding introduces a heightened level of complexity, defying straightforward categorization. In our research, we delineate the notion of continuous tasks based on distinct semantic contents, such as "human", "animal" and "food". Besides, we follow the class- and domain-incremental settings, where the identity of the task remains unknown during test time, posing a more common and formidable challenge.

## 3.2 Base Framework for Continual VL Retrieval and Grounding

In this section, we first build a simple but effective framework with existing prompt-based approaches [48, 50] to serve as our baseline for continuous vision-language retrieval and grounding.

In the base framework, we adopt the pre-trained multi-modal transformer model, such as the CLIP [38] or GLIP [25], as the feature extractor. These pre-trained transformer can be simplified to consist of a language encoder $\phi_q(\cdot)$ and a vision encoder $\phi_v(\cdot)$, while the rest components such as interaction module are omitted here for clarity. Throughout the training process, we keep these models frozen to preserve their learned representations. To tailor these models for downstream vision-language applications, we introduce a trainable downstream head $H(\cdot)$ on top of the pre-trained transformer. The head is designed to be relatively lightweight to fully leverage the potential of large pre-trained model. Concretely, for image-text retrieval, the head comprises a single MLP layer for each encoder output, as the feature similarity can be directly calculated. For video-text retrieval, the head further incorporates a mean-pooling layer on the vision side to aggregate the clip-level features. For referring expression comprehension and segmentation,

we follow [26] to build the head as a simple multi-modal transformer model with a one-layer encoder, two-layer decoders, and an MLP layer to obtain center coordinates or several fully-connected convolutional layers to derive segmentation masks.

**Prompt Design.** Following previous works [48], for each task $T^k$, we use an independent set of continuous learnable parameters $P^k \in \mathbb{R}^{L \times D}$ as a part of inputs to the pre-trained encoder, where $L \in \mathbb{R}$ and $D \in \mathbb{R}$ indicate the prompt's length and embedding dimension respectively. To incorporate the prompt, we extend the embedding of any single modality input from task $T^k$ as $\bar{x} = [x, P^k]$, where $x \in \{q, v\}$ denotes the original input tokens of the vision or language modality. This extended embedding is then fed into the transformer blocks. When trained on a new task $T_{k+1}$, a new independent set of prompts $P^{k+1}$ is added. Hence, learning all the tasks sequentially results in a task-wise prompt pool $P = \{P^1, P^2, ..., P^K\}$, where $P^k \in \mathbb{R}^{L \times D}$ is a single set of prompts of task $T^k$, and $K$ is the total number of tasks. Since we have separate encoders for vision and language, we also build two modality-specific prompt pools $P^v = \{P^{v,k}\}_{k=1}^K$ and $P^q = \{P^{q,k}\}_{k=1}^K$.

**Downstream Training.** Given the extended embedding $\bar{q}$ and $\bar{v}$, we computed encoded features by $\tilde{q} = \phi_q(\bar{q})$ and $\tilde{v} = \phi_v(\bar{v})$. Then we input these features into the downstream head and obtain the problem-specific output $o$, where $o = H(\tilde{q}, \tilde{v})$. At task $T^k$, we develop the problem-specific loss $\mathcal{L}_{base} = \mathcal{L}_{problem}(o^k, a^k)$, where $\mathcal{L}_{problem}$ corresponds to the contrastive loss for two retrieval tasks [38], the L1 loss for referring expression comprehension and the dice loss for referring expression segmentation [25]. During training, only the parameters of the prompt and the downstream head are updated while the remaining parameter are kept frozen.

**Inference.** During inference, since we learn independent prompt groups for each task, we need to identify which prompt group should be leveraged for the input test data. Here we follow S-Prompts [48] to apply K-Means to store the data centriods of each task during training, and use K-NN to search for the nearest centroid of the given test feature, so as to identify its corresponding prompt group during inference. This simple strategy can work generally well in our experiments.

## 3.3 Intra-modal Relevance Estimation

As shown in Figure 2, we introduce intra-modal relevance estimation (IRE) for both vision and language encoders, where we leverage the task relevance within each modality to optimize prompts.

The objective of IRE is to learn a binary classifier to judge whether the input prompt is relevant or irrelevant to the data of current task $T^k$. Inspired by BERT [9], we insert a special token [REL] to obtain the extended embedding as $\bar{x} = [x, P^k, [REL]]$, where $x \in \{q, v\}$. Then we input $\bar{x}$ to the transformer encoder and obtain the encoded features $\tilde{x} \in \{\tilde{q}, \tilde{v}\}$. By adding a linear layer on the output corresponding to the token [REL], we calculate a scalar representing the logits $R$. To estimate the relevance, we regard the prompt $P^k$ that corresponds to the current $T^k$ as the positive instances (i.e. relevant pairs) and construct negative instances (i.e. irrelevant pairs) by randomly substituting $P^k$ with one of the historical prompts $P^i$ from past tasks $T^i$, where $0 \le i < k$. The full intra-modal relevance estimation is performed using a binary-cross

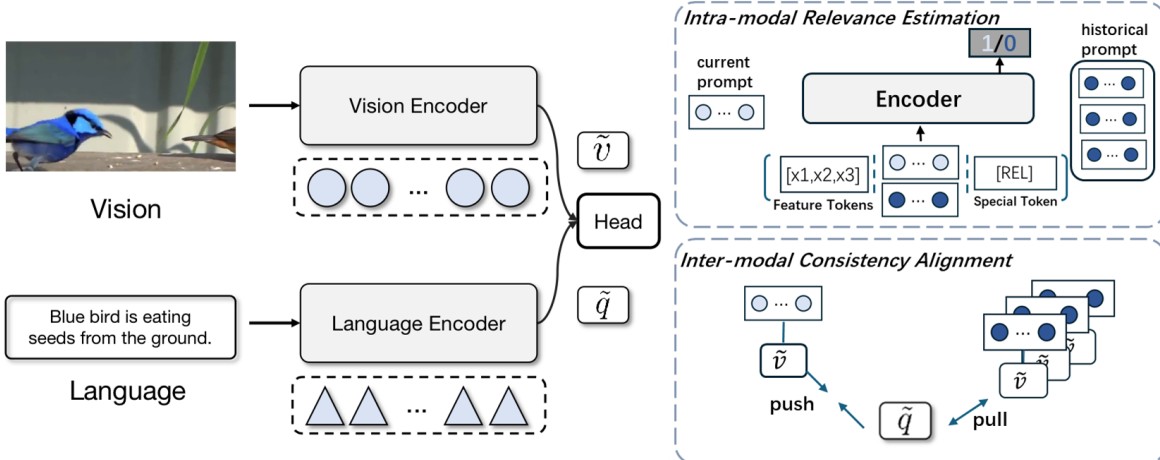

**Figure 2: The framework of Historical Prompt Calibration.**

entropy loss:

$$\mathcal{L}_{\text{IRE}} = -\log R^+ - \log(1 - R^-), \tag{1}$$

where $R^+$ and $R^-$ are the logits of relevant and irrelevant pairs respectively. As we employ two distinct encoders for the vision and language modalities, we perform relevance estimation separately for each modality.

For the initial task $T^1$, as no previous tasks exist, we only apply the loss $\mathcal{L}_{\text{base}}$. As we progress to the second task $T^2$, we introduce the IRE loss $\mathcal{L}_{\text{IRE}}$ and update both the special token [REL] and the linear layer. For subsequent tasks $T^i$ where $i > 2$, we fix the special token [REL] and update the linear layer with a momentum way. This step is key to learn effective prompt representations that contain much task-specific information, since the relevance estimation also suffers the *catastrophic forgetting* problem and continuous updating the token or linear layer will lead to severe forgetting of previous relevance knowledge. Instead, we fix the token to preserve the initial relevance information and slowly update the linear layer to avoid rapid forgetting. In this process, the prompt parameters of the current task can be gradually distinctive from the previous ones and therefore be more task-specific.

## 3.4 Inter-modal Consistency Alignment

We design inter-modal consistency alignment (ICA) to uniformly guide the optimization of prompts across modalities, as illustrated in Figure 2.

The objective of ICA is to perform contrastive learning over the encoded cross-modal features generated from current and historical prompts. Given the input data $v^k, q^k$ and the prompt $P^{v,k}, P^{i,k}$ that are associated with the current task $T^k$, we can calculate the encoded features $\tilde{v}^k, \tilde{q}^k$ and utilize the problem-specific loss $\mathcal{L}_{\text{base}}$ to learn cross-modal alignment. However, relying solely on the data from the current task during training can limit the potential for achieving optimal alignment between the features $\tilde{v}$ and $\tilde{q}$, as presented in 1.

Hence, we adopt the contrastive scheme by integrating the historical prompts to construct negative pairs, which can regularize

the learning for current prompts. Concretely, given $\tilde{v}^k$ and $\tilde{q}^k$ as the positive pair, we can generate $2 \times (k-1)$ negative instances $\{\tilde{v}^{i,-}\}_{i=1}^{k-1}$ and $\{\tilde{q}^{i,-}\}_{i=1}^{k-1}$ by replacing the prompt $P^{v,k}$ and $P^{q,k}$ with the previous prompts $P^{v,i}$ and $P^{q,i}$ where $i < k$. Then we develop a contrastive alignment loss to pull each modal feature $\tilde{v}^k$ and $\tilde{q}^k$ close to each other while push them apart from all other negative instances $\{\tilde{q}^{i,-}\}_{i=1}^{k-1}$ and $\{\tilde{v}^{i,-}\}_{i=1}^{k-1}$. On the language side, the loss can be given by:

$$\mathcal{L}_{\text{ICA}}^{\text{lang}} = -\log \frac{\exp(\tilde{q}^k \cdot \tilde{v}^k / \tau)}{\exp(\tilde{q}^k \cdot \tilde{v}^k / \tau) + \sum_{i=1}^{k-1} \exp(\tilde{q}^t \cdot \tilde{v}^{i,-} / \tau)}, \tag{2}$$

where $\tau$ is a temperature parameter [13] set to 0.1. The loss $\mathcal{L}_{\text{ICA}}^{\text{vis}}$ on the vision side can be obtain similarly. The full loss $\mathcal{L}_{\text{ICA}} = \mathcal{L}_{\text{ICA}}^{\text{lang}} + \mathcal{L}_{\text{ICA}}^{\text{vis}}$.

The training process of the inter-modal consistency alignment follows a similar pattern to intra-modal relevance estimation, which starts from the second task as it requires the preparation of historical tasks. By utilizing the same modal features with prompts of different tasks, our constructed contrastive samples can more effectively enhance the consistency of cross-modal features, and ensure that this enhancement comes from the prompt corresponding to the task, thereby calibrating the cohesive optimization direction of the prompt.

## 3.5 Training

We combine the problem-specific loss and two proposed calibration loss to train the model, i.e.,

$$\mathcal{L}_{\text{HPC}} = \lambda_1 \mathcal{L}_{\text{base}} + \lambda_2 \mathcal{L}_{\text{IRE}} + \lambda_3 \mathcal{L}_{\text{ICA}}, \tag{3}$$

where $\lambda_1$, $\lambda_2$ and $\lambda_3$ are set to 1.0, 0.1, 0.1 to balance the three losses. As our above discussion, we introduce two calibration losses exclusively after the initial task. It is important to note that our HPC paradigm is solely incorporated into the training process, ensuring that it has no impact on the inference speed or runtime efficiency.

# 4 EXPERIMENTS

In this section, we elaborate the experiment setting, performance evaluation of the proposed method, ablation study, in-depth analysis and hyper-parameter analysis. Additional dataset details and implementation details are provided in the supplementary materials.

## 4.1 Experiment Setting

**Task and Dataset.** We perform experiments on four mainstream vision-language tasks, including two retrieval tasks and two grounding tasks:

- Image-text Retrieval: We select MS-COCO [29] for evaluation and divide it into 12 tasks based on content categories, e.g. person, vehicle, animal, etc.
- Video-text Retrieval: We select MSR-VTT [52] for evaluation and divide into 10 tasks based on content categories, e.g. news, movie, sports, etc.
- Referring Expression Comprehension: We select RefCOCO [54] for evaluation and divide into 12 tasks based on content categories as MS-COCO.
- Referring Expression Segmentation: Similar to the comprehension task, we select RefCOCO [54] for evaluation and divide into the same 12 tasks.

The former two tasks focus on global-level alignment while the latter two are more difficult with local-level grounding. To generate separate tasks for continual learning for the image-text retrieval, we use the "category" annotation in original dataset to define 12 tasks, as shown in Table 1. To generate separate tasks for continual learning for the video-text retrieval, we use the "category" annotation in original dataset to define 10 tasks, as shown in Table 2. We evaluate the referring expression comprehension and segmentation on this dataset via its box-level and mask-level annotations respectively. Similar to MS-COCO, we adopt the same division split in Table 1.

**Table 1: Task division in Coco-based datasets.**

| Person | Vehicle | Outdoor | Animal | Accessory | Sport |
|---|---|---|---|---|---|
| Kitchen | Food | Furniture | Electronic | Appliance | Indoor |

**Table 2: Task division in MSR-VTT dataset.**

| News | Movie | Sports | Cooking | Traffic |
|---|---|---|---|---|
| Animation | Music | Animal | Kids | Beauty |

**Metric.** Under the settings where the task boundaries are unknown and each task has an associated test set, we follow previous works [33, 50] to adopt two widely-used metrics, i.e., average accuracy and forgetting. Specifically, for two retrieval tasks, we report the accuracy as R@K (Recall at K) that calculates the percentage of test samples for which the correct result is found in the top-K retrieved points to the query sample. For two grounding tasks, we compute the accuracy as mIoU (Mean Intersection-over-Union) by the averaging over the IoU of each testing sample where IoU is the intersection area divided by the total union area. To better reflect the computational cost of methods, we also report the number of parameters that need to be tuned.

**Implementation Details.** For input data, all the training images or video frames are resized to 240 x 240 and 320 x 320 for retrieval

and grounding tasks, respectively. For model selection, we follow the prompt-based pipeline to select two powerful and widely-used pre-trained transformer models as encoders: CLIP(ViT-B/16) [38] for retrieval tasks and GLIP-T [25] for grounding tasks. For prompt design, we adopt the deep prompt [18] to insert prompt for all encoder layers. In CLIP, the prompt length is set to 16 and 12 for language and vision encoders, respectively. In GLIP, the prompt length is set to 12 and 20 for language and vision encoders, respectively. To train our model, we use the Adam optimizer with the cosine scheduler.

We adopt Adam optimizer with a momentum of 0.9 and a cosine annealing scheduler [34]. For image-text retrieval, we train the model for 5 epochs for each task. The batch size is 128. The learning rate is set to 0.05 and 0.01 for prompt tokens and the head, respectively. For video-text retrieval, we train the model for 10 epochs for each task. The batch size is 24. The learning rate is set to 0.01 and 0.0001 for prompt tokens and the head, respectively. The head structure is the same as [36]. For referring expression comprehension and segmentation, we train the model for 20 and 10 epochs for each task, respectively. The batch size is 16. The learning rate is set to 0.001 and 0.0001 for prompt tokens and the head, respectively. The head structure is the same as [26].

## 4.2 Performance Evaluation

**Methods.** We compare the proposed HPC strategy with the following state-of-the-art methods for continual learning, which can be divided into three groups: (1) Replay-based method. The replay-head is the naive sequential fine-tuning approach with the pre-trained model frozen. The replay-all instead fine-tunes pre-trained model weights as well. (2) Regularization-based methods including EWC [22], LwF [28]. (3) Prompt-based methods including DyTox [12], L2P [50], S-Prompts [48], DualPrompt [49], MaPLe [21] and DCP [31]. Specifically, L2P and S-Prompts extract features using the uni-encoder, whereas DualPrompt, MaPLe and DCP take into account the interactions between modalities during feature extraction. As these methods are designed for uni-modal application, we extend them by separately applying their strategy on two modal encoders. To compare fairly, we use the same pre-trained model (i.e., CLIP/GLIP) for all compared methods as well as ours. The full implementations details of baselines are shown in the appendix.

**Baseline Implementation.** For all baselines, we adopt the same base network as our HPC that consists of pre-trained CLIP or GLIP. (1) For replay-based methods, we simply maintain a extra memory space to store the training samples from the previous tasks, where the memory size is set to 50 or 100 samples per task. (2) For regularization-based methods EWC [22] and LwF [28], we adopt the same hyper-parameter setting as reported in their work. (3) For prompt-based methods: In DyTox [12], we use 3 Self-Attention Blocks and 1 Task-Attention Block for retrieval, and 1 Self-Attention Blocks and 1 Task-Attention Block for grounding, where all have 8 attention heads. To transfer the task-specific classifier in original classification problem, we also design task-specific structure by keeping separate MLP layers for different tasks in retrieval and separate MLP layer/fully-convolutional layers for different tasks in referring expression comprehension/segmentation. In L2P [50], we set the prompt pool size to 20/10, and the number of key selection

**Table 3: Performance Evaluation on Image-Text Retrieval Task. Bold: best results of exemplar-free methods, Underline: second best results of exemplar-free methods. Upper-bound: supervised finetuning on the i.i.d. data of all tasks, which is usually regarded as the upper bound performance[50].**

| Method | Buffer size | Text Retrieval | | | Image Retrieval | | | Param (↓) |
|---|---|---|---|---|---|---|---|---|
| | | R@1 (↑) | R@5 (↑) | Forget(↓) | R@1 (↑) | R@5 (↑) | Forget(↓) | |
| Replay-all | 50/class | 36.24 | 55.49 | 32.65 | 29.72 | 55.28 | 30.35 | 100% |
| Replay-all | 100/class | 43.51 | 63.21 | 24.19 | 34.54 | 62.61 | 24.43 | 100% |
| Replay-head | 50/class | 39.60 | 58.82 | 28.59 | 35.04 | 60.68 | 23.12 | 0.25% |
| Replay-head | 100/class | 48.24 | 66.39 | 15.46 | 36.39 | 64.39 | 14.80 | 0.25% |
| EWC [22] | 0/class | 29.24 | 41.60 | 31.08 | 29.36 | 53.56 | 24.71 | 0.25% |
| LWF [28] | | 32.19 | 50.06 | 27.44 | 32.01 | 58.37 | 21.20 | 0.25% |
| DyTox [12] | | 49.83 | 70.64 | 20.13 | 38.54 | 63.79 | 16.87 | 5.68% |
| L2P [50] | | 57.94 | 78.26 | 12.14 | 43.17 | 69.38 | 10.39 | 1.94% |
| S-Prompts [48] | 0/class | 63.90 | 87.95 | 6.52 | 49.82 | 78.85 | 5.73 | 5.36% |
| DualPrompt [49] | | 64.48 | 88.09 | 5.89 | 50.39 | 79.12 | 5.04 | 2.71% |
| MaPLe [21] | | 64.92 | 89.27 | 6.04 | 51.67 | 80.83 | 5.79 | 7.43% |
| DCP [31] | | 64.03 | 89.56 | 6.22 | 50.21 | 81.14 | 6.13 | 10.35% |
| **HPC** (ours) | | **65.94** | **90.51** | **4.21** | **52.61** | **82.03** | **3.86** | 2.72% |
| Upper-bound | | 68.19 | 92.71 | - | 54.83 | 85.54 | - | 2.72% |

**Table 4: Performance Evaluation on Video-Text Retrieval Task. Bold: best results of exemplar-free methods, Underline: second best results of exemplar-free methods. Upper-bound: supervised finetuning on the i.i.d. data of all tasks, which is usually regarded as the upper bound performance[50].**

| Method | Buffer size | Text Retrieval | | | Video Retrieval | | | Param (↓) |
|---|---|---|---|---|---|---|---|---|
| | | R@1 (↑) | R@5 (↑) | Forget(↓) | R@1 (↑) | R@5 (↑) | Forget(↓) | |
| Replay-all | 50/class | 22.15 | 42.23 | 23.06 | 24.38 | 44.92 | 25.17 | 100% |
| Replay-all | 100/class | 24.20 | 44.73 | 20.49 | 25.26 | 46.11 | 22.36 | 100% |
| Replay-head | 50/class | 26.13 | 50.33 | 12.20 | 30.41 | 52.62 | 12.42 | 0.25% |
| Replay-head | 100/class | 28.03 | 54.28 | 9.84 | 32.13 | 55.39 | 10.11 | 0.25% |
| EWC [22] | 0/class | 17.66 | 32.49 | 16.26 | 18.43 | 34.51 | 17.44 | 0.25% |
| LWF [28] | | 22.57 | 41.32 | 11.38 | 23.64 | 42.58 | 12.96 | 0.25% |
| DyTox [12] | | 30.34 | 57.03 | 11.08 | 34.22 | 58.90 | 11.44 | 5.68% |
| L2P [50] | | 34.80 | 59.62 | 9.25 | 37.24 | 62.56 | 9.63 | 1.94% |
| S-Prompts [48] | 0/class | 36.41 | 64.33 | 7.39 | 39.15 | 66.72 | 8.14 | 5.36% |
| DualPrompt [49] | | 38.02 | 66.21 | 6.63 | 40.72 | 69.38 | 6.44 | 2.71% |
| MaPLe [21] | | 38.92 | 66.43 | 5.21 | 42.15 | 70.28 | 5.83 | 7.43% |
| DCP [48] | | 38.08 | 66.87 | 5.64 | 41.58 | 69.14 | 7.59 | 10.35% |
| **HPC** (ours) | | **39.41** | **67.56** | **4.92** | **42.39** | **71.49** | **5.11** | 2.72% |
| Upper-bound | - | 42.74 | 70.39 | - | 45.60 | 74.85 | - | 2.72% |

to 5/5 for vision and language modality. For DualPrompt [49], the sharable prompt length is set to 16 and 12 for language and vision encoders in CLIP, respectively, and is set to 6 and 10 for language and vision encoders in GLIP. For S-Prompts [48], MaPLe [21], and DCP [31] we adopt the same parameter setting as our HPC, i.e. the prompt length is set to 16 and 12 for language and vision encoders in CLIP, respectively; the prompt length is set to 12 and 20 for language and vision encoders in GLIP, respectively.

**Results of Retrieval Tasks.** The results of two retrieval tasks are shown in Table 3 and 4. It demonstrates that our proposed HPC method significantly outperforms the other exemplar-free methods, e.g. S-Prompts, DualPrompt. HPC slightly outperforms MaPLe and DCP with fewer additional parameters. We also find the proposed HPC methods' forgetting degrees are much less than those of the others. An interesting point is that the replay-based methods achieve limited performance. We argue the reason is that the limited data in each task restrict the retrieval learning since

**Table 5: Performance Evaluation on Two Vision-Language Grounding Tasks. Bold: best results of exemplar-free methods, Underline: second best results of exemplar-free methods. Upper-bound: supervised finetuning of HPC on the i.i.d. data of all tasks, which is usually regarded as the upper bound performance[50].**

| Method | Buffer size | Referring Expression Comprehension | | | Referring Expression Segmentation | | |
|---|---|---|---|---|---|---|---|
| | | mIoU ($\uparrow$) | Forget ($\downarrow$) | Param ($\downarrow$) | mIoU ($\uparrow$) | Forget ($\downarrow$) | Param ($\downarrow$) |
| Replay-all | 50/class | 64.60 | 14.39 | 100% | 56.37 | 12.16 | 100% |
| Replay-all | 100/class | 68.71 | 8.34 | 100% | 60.05 | 8.83 | 100% |
| Replay-head | 50/class | 40.19 | 10.35 | 2.39% | 42.18 | 11.82 | 3.61% |
| Replay-head | 100/class | 42.35 | 9.64 | 2.39% | 44.24 | 10.39 | 3.61% |
| EWC [22] | 0/class | 45.83 | 23.89 | 2.39% | 48.30 | 26.12 | 3.61% |
| LWF [28] | | 51.06 | 17.56 | 2.39% | 53.74 | 21.33 | 3.61% |
| DyTox [12] | | 57.53 | 19.81 | 3.55% | 50.67 | 18.61 | 5.71% |
| L2P [50] | | 64.46 | 11.02 | 2.59% | 55.37 | 13.55 | 3.80% |
| S-Prompts [48] | 0/class | 68.54 | _7.32_ | 3.39% | 60.17 | _8.70_ | 5.58% |
| DualPrompt [49] | | 66.39 | 10.74 | 2.82% | 59.51 | 10.14 | 4.03% |
| MaPLe [21] | | _69.81_ | 8.17 | 8.54% | _63.37_ | 8.96 | 10.23% |
| DCP [31] | | 67.49 | 9.72 | 10.28% | 62.21 | 9.45 | 12.56% |
| **HPC** (ours) | | **71.04** | **5.36** | 3.04% | **64.32** | **5.89** | 4.25% |
| Upper-bound | - | 78.35 | - | 3.04% | 70.56 | - | 4.25% |

**Table 6: Ablation studies of our Historical Prompt Calibration method.**

| Task | Metrics | **HPC** | HPC(w/o. IRE) | HPC(w/o. ICA) | HPC(IRE w/o. estimator) | HPC(ICA w/o. contrastive) |
|---|---|---|---|---|---|---|
| Image-Text Retrieval | TextR@1 $\uparrow$ | **65.94** | 64.79 | 64.13 | 65.03 | 64.22 |
| | Forget $\downarrow$ | **4.21** | 4.89 | 5.35 | 4.64 | 5.05 |
| Referring Image Segmentation | mIoU $\uparrow$ | **64.32** | 63.24 | 62.51 | 63.28 | 61.89 |
| | Forget $\downarrow$ | **5.89** | 6.73 | 7.60 | 6.94 | 8.24 |

**Table 7: Performance Comparison on Grounding Tasks with Complex Downstream Head. The value in bracket denotes the original results.**

| Method | Referring Expression Comprehension | | Referring Expression Segmentation | |
|---|---|---|---|---|
| | mIoU | Param | mIoU | Param |
| L2P [50] | $63.33_{(64.46)}$ | $7.98_{(2.39)}$ | $53.60_{(55.37)}$ | $9.07_{(3.80)}$ |
| DualPrompt [49] | $64.71_{(66.39)}$ | $8.20_{(2.82)}$ | $56.65_{(59.51)}$ | $9.28_{(4.03)}$ |
| **HPC** (ours) | $\mathbf{70.83}_{(71.04)}$ | $8.36_{(2.82)}$ | $\mathbf{64.16}_{(64.32)}$ | $9.44_{(4.25)}$ |

it requires large contrastive samples, thereby making the model biased. This can explain the fact that freezing the pre-trained model and only tune the head can bring better performance. Instead, our method only fine-tune the prompt parameter and avoid wasting the pre-trained model's knowledge. The inferior performance of other prompt-based methods might be due to the less transfer learning capability of their dependent prompt learning on heterogeneous tasks and modalities.

**Results of Grounding Tasks.** The results of two challenging grounding tasks are summarized in Table 5. We can find HPC still surpass most exemplar-free methods. Different from the retrieval, we observe the replay-based method is effective and tuning all model can bring further gains. While these methods can reach closer to our method, it requests a large memory overhead that

increases linearly with the class number for the storage of class-wise exemplars and it also requires a large amount of parameters for tuning. Besides, we notice our method brings more remarkable gains in segmentation than comprehension. We argue the reason is that segmentation is more difficult and designed with complex downstream head, which indirectly poses a greater obstacle for continual learning. This fact again validates the superiority of our method.

## 4.3 Ablation Study

We conduct ablation studies of Calibrate Prompt and show the results in Table 6. First, we develop two overall ablated methods HPC (w/o. IRE) and HPC (w/o. ICA) by discarding the intra-modal

relevance estimation and inter-modal consistency alignment respectively. The results demonstrate that the two calibration tasks can both help improve the model performance in continuous tasks, showing our method can effectively calibrate prompt for the pre-trained transformer.

Besides, we further explore detailed design by generating two ablation methods HPC (IRE w/o. estimator) and HPC (ICA w/o. contrastive). The HPC (IRE w/o. estimator) leverages the contrastive loss to enlarge the gap between prompt groups of different tasks instead of our estimator on the transformer model. The HPC (ICA w/o. contrastive) directly enhances the cross-modal alignment without contrasting to historical prompts. The results indicate that our design is effective, since the estimator can leverage the discriminative power of pre-trained model and the contrastive loss can better achieve cross-modal alignment by utilizing historical information.

## 4.4 In-depth Analysis

**Sensitivity to Downstream Head.** We validate the robustness of our method by replacing the lightweight task-specific head with more complex one. We consider the challenging grounding tasks and follow [26] to build a 12 layer multi-modal transformer architecture and introduce cross-modal interaction mechanism. From the results shown in Table 7, we can clearly observe that all methods suffer performance degradation since the complex neural network of downstream head lead to more severe forgetting. However, our method still outperforms all other methods, validating it can promote the pre-trained encoder to obtain more generalizable features for even complex downstream structure and alleviate the latent forgetting of it.

**Performance with Task Number.** To study the correlation between the effect of our method and the number of tasks, we visualize the accuracy curves with the incremental tasks in Figure 3. It can be seen that the gap between our HPC method and other methods continues to grow with the input of the task. This fact is because that our intra-modal relevance estimation and inter-modal consistency alignment can leverage the historical information of previous tasks to improve the learning of current and upcoming tasks. Compared with S-Prompts that builds independent prompt pools, our method can explicitly utilize the knowledge of each past task, thus bringing a more direct improvement.

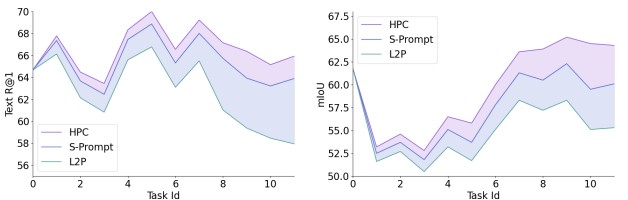

(a) Image-Text Retrieval  (b) Referring Expression Segmentation

Figure 3: Performance with Task Number.

**Effectiveness of the temperature parameter.** We explore the effect of the temperature parameter $\tau$, which is a crucial hyper-parameter in loss $\mathcal{L}_{ICA}$ of inter-model consistency alignment. We set the value of $\tau$ to $[0.01, 0.1, 0.2, 0.5, 1.0]$ and display the results in

Figure 4. We observe that the performance of our method decreases when the temperature is too small or too large. The optimal temperature value is around 0.1. This indicates a proper temperature value can promote feature learning, which is consistent with previous studies [13].

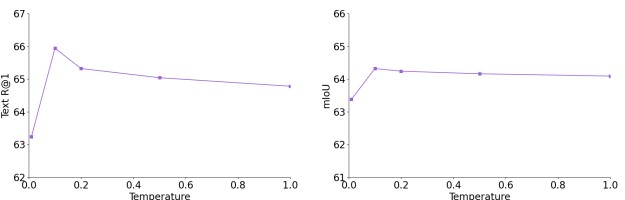

(a) Image-Text Retrieval  (b) Referring Expression Segmentation

Figure 4: Impact of Temperature Parameter $\tau$.

**Training Stability.** Further, we draw the three loss curve that corresponds to downstream task, IRE and ICA respectively during training. It is visually apparent in Figure 5 that the two losses $\mathcal{L}_{IRE}$ and $\mathcal{L}_{ICA}$ converge at a faster rate than the primary task loss $\mathcal{L}_{base}$, reaching its order of magnitude within one percent. Hence, our proposed strategy will not impede the training of the primary task, but instead enables prompt calibration in the early training stage, leading to more effective prompt representations for continual learning.

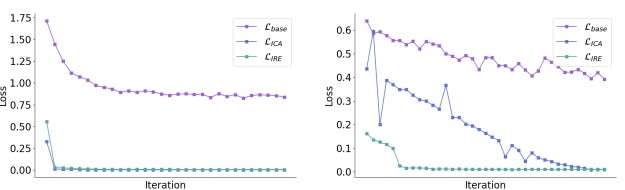

(a) Image-Text Retrieval  (b) Referring Expression Segmentation

Figure 5: Training loss curve of our HPC method.

## 5 CONCLUSIONS

In this work, we investigate the continual learning for vision-language retrieval and grounding tasks. Inspired by the recent success of pre-trained multi-modal transformer and the prompt technique, we adopt the prompt-based pipeline to solve this problem. To solve two critical limitations, i.e. the unfamiliarity across tasks and the heterogeneity between modalities, we propose Historical Prompt Calibration. First, we design a intra-modal relevance estimation task to help prompt encode more task-specific information by distinguishing the previous prompts of historical tasks, which is achieved by a relevance estimator on top of the pre-trained encoder. Second, we develop a inter-modal consistency alignment training to enhance cross-modal alignment by contrasting one modal features with another modal features, where contrastive samples are generated by prompts of current and previous tasks. In two vision-language retrieval tasks and two vision-language grounding tasks, HPC shows superior performance over the state-of-the-art methods.

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
