# OpenReview forum: "Calibrating Prompt from History for Continual Vision-Language Retrieval and Grounding"
_acmmm.org/ACMMM/2024/Conference — MM2024 Poster_

### Official Review · Reviewer_jNqW · 2024-04-29

**Rating:** 4
**Confidence:** 3

**Summary:**

This paper introduces a novel continual learning paradigm tailored for two vision-language applications: retrieval and grounding. Specifically, the authors propose a method termed Historical Prompt Calibration, which calibrates prompts in two key ways. Firstly, it involves intra-modal relevance estimation that encodes task-specific information. Secondly, it features inter-modal consistency alignment to enhance the agreement between modality-specific prompts. Extensive experiments across four vision-language tasks—namely, image-text retrieval, video-text retrieval, referring expression comprehension, and segmentation—demonstrate the effectiveness of the proposed method. This approach not only addresses the complexities inherent in cross-modal interactions but also shows promising improvements in performance across diverse tasks.

**Strengths:**

1. $\textbf{Innovative Approach}$: This manuscript introduces Historical Prompt Calibration (HPC), a novel strategy for continual learning in vision-language tasks. This method addresses critical limitations in existing prompt-based methods by introducing intra-modal relevance estimation and inter-modal consistency alignment, which enhance task-specific information encoding and cross-modal alignment, respectively.
2. $\textbf{Logical Writing}$: The manuscript is well-structured, beginning with a clear introduction that outlines the limitations of existing methods and logically progresses to present the proposed HPC approach. Each section flows coherently, making it easy for the reader to follow the development of ideas and understand the contributions of the work.

**Limitations:**

1. $\textbf{Lacking of Innovation}$: While the Historical Prompt Calibration (HPC) strategy introduces intra-modal relevance estimation and inter-modal consistency alignment, these techniques may be seen as incremental improvements rather than groundbreaking innovations. The core ideas are extensions of existing prompt calibration and continual learning methods, which are quite incremental.
2. $\textbf{Lacking of Discussions}: While the manuscript emphasizes the effectiveness of Historical Prompt Calibration (HPC), it lacks a detailed discussion on the computational overheads introduced by the intra-modal relevance estimation and inter-modal consistency alignment processes. Understanding the trade-offs between improved performance and computational complexity is crucial for practical applications.

**Suitability:**

3

---

### Official Review · Reviewer_EPmK · 2024-05-21

**Rating:** 4
**Confidence:** 3

**Summary:**

This paper explores continual learning for vision-language applications, focusing on retrieval and grounding tasks, and proposes a novel strategy called Historical Prompt Calibration to address challenges related to task unfamiliarity and modality heterogeneity by leveraging pre-trained transformer models and prompt techniques.

**Strengths:**

The paper introduces the concept of Historical Prompt Calibration (HPC), which is a novel approach to addressing the challenges of continual learning in vision-language tasks.

**Limitations:**

W1: In the experiments conducted in this paper, the methods compared are mostly those of few-shot VLM adaptation, and the comparison with methods for Continual Learning for retrieval and localization is relatively outdated. The authors should compare with newer methods, i.e., [1][2]. However, in addition to the suggestions mentioned above, the authors could also reference relevant work from the past two years.

W2: Formatting Issue. The authors should ensure that the text in the figures and tables matches that in the main text, for example, in Figure 5.


[1] Continual Vision-Language Retrieval via Dynamic Knowledge Rectification. AAAI, 2024.
[2] Boosting Continual Learning of Vision-Language Models via Mixture-of-Experts Adapters. CVPR, 2024.

**Suitability:**

3

---

### Official Review · Reviewer_9wcH · 2024-05-24

**Rating:** 4
**Confidence:** 3

**Summary:**

This article presents an approach, Historical Prompt Calibration (HPC). This approach is used to address the problems of task unfamiliarity and inter-modal heterogeneity that arise in multimodal continuous learning. For task unfamiliarity, HPC utilizes cues from past tasks to improve the relevance between the current task and the cues. For the inter-modal heterogeneity problem, this paper proposes to use contrast learning to achieve inter-modal consistency alignment, which in turn enhances the consistency between cues from two modalities under the same task. Experiments on four visual-verbal tasks demonstrate the effectiveness and superiority of HPC.

**Strengths:**

1. The paper designs a reasonable experimental protocol for this paper and implements a Baseline for all the current better-performing continuous learning scenarios to achieve a fair comparison in the context of cross-modal continuous learning.
2. The experiments in 4.4 on downstream headers, temperature coefficients, and training stability are persuasive.

**Limitations:**

1. In Figure 5, IRE and ICA achieve the calibration of cues early in the training, is it possible to explore the performance performance and knowledge forgetting of HPC doing continuous learning in four tasks in the few-shot case?
2. Could you provide some visualized grounding results?
3. The experiments under the text-image retrieval, and text-video retrieval tasks in this paper are performed by splitting the COCO dataset and using a series of subsets of the COCO dataset as data streams. In fact, the data stream can also be constructed with many different datasets. The differences in image features between multiple different datasets may be greater than using a subset of the same dataset, and the challenge of catastrophic forgetting during continuous learning may be more difficult. Some recent work on continuous learning across modalities [1] has used multiple different datasets as data streams to conduct experiments, could this paper also use this approach to construct data streams to conduct experiments?

Reference
[1] Zangwei Zheng, Mingyuan Ma, Kai Wang, Ziheng Qin, Xiangyu Yue, and Yang You. Preventing zero-shot transfer degradation in continual learning of vision-language models. arXiv preprint arXiv:2303.06628, 2023.

**Suitability:**

2

---

### Official Review · Reviewer_fRbX · 2024-05-27

**Rating:** 4
**Confidence:** 4

**Summary:**

The paper investigates the use of continual learning in multi-modal settings, focusing on vision-language applications such as retrieval and grounding. The authors identify two critical limitations in existing methods: unfamiliarity across tasks and heterogeneity between modalities. To address these issues, they propose Historical Prompt Calibration, which includes intra-modal relevance estimation and inter-modal consistency alignment. The proposed approach is evaluated on four vision-language applications, demonstrating its superiority over state-of-the-art methods.

**Strengths:**

- The use of pre-trained transformer models (CLIP/GLIP) and prompt techniques to address the identified limitations is well-grounded in theory.

- The proposed objectives (intra-modal relevance estimation and inter-modal consistency alignment) are logically sound and effectively address the identified issues.

- The findings have significant implications for improving vision-language models, making them more adaptable to real-world, non-stationary data distributions.

**Limitations:**

- Although the paper claims superiority over state-of-the-art methods, a more detailed comparative analysis with a diverse set of baseline models would strengthen the findings.

- The approach might require task-specific adjustments for optimal performance, which could limit its applicability in a wider range of scenarios without further tuning. This suggests that while the method is effective for the evaluated tasks, it may not be readily applicable to other tasks without significant modifications.

- The reliance on pre-trained transformer models like CLIP/GLIP may require significant computational resources, which could be a barrier for some applications. Additionally, the environmental impact of extensive computational requirements should be considered.

- The paper fails to cite many important prior works.

1. Oh, Changdae, et al. "Towards calibrated robust fine-tuning of vision-language models." NeurIPS Workshop on Distribution Shifts. 2023.

2. Cheng, Zhi-Qi, et al. "Gsrformer: Grounded situation recognition transformer with alternate semantic attention refinement." Proceedings of the 30th ACM International Conference on Multimedia. 2022.

3. Fang, Kaipeng, et al. "ProS: Prompting-to-simulate Generalized knowledge for Universal Cross-Domain Retrieval." Proceedings of the IEEE/CVF Conference on Computer Vision and Pattern Recognition. 2024.

**Suitability:**

2

---

### Meta-Review · Area_Chair_PEpi · 2024-06-28

**Recommendation:** Accept (Poster)
**Confidence:** 5

**Metareview:**

All reviewers found the proposed method to be novel and effective.  The authors are encouraged to improve the final version by following reviewer recommendations.